# Magnetically Guided Localization Using a Guiding-Marker System^®^ and a Handheld Magnetic Probe for Nonpalpable Breast Lesions: A Multicenter Feasibility Study in Japan

**DOI:** 10.3390/cancers13122923

**Published:** 2021-06-11

**Authors:** Tomoko Kurita, Kanae Taruno, Seigo Nakamura, Hiroyuki Takei, Katsutoshi Enokido, Takashi Kuwayama, Yoko Kanada, Sadako Akashi-Tanaka, Misaki Matsuyanagi, Meishi Hankyo, Keiko Yanagihara, Takashi Sakatani, Kentaro Sakamaki, Akihiro Kuwahata, Masaki Sekino, Moriaki Kusakabe

**Affiliations:** 1Department of Breast Surgery and Oncology, Nippon Medical School Hospital, Tokyo 113-8603, Japan; takei-hiroyuki@nms.ac.jp (H.T.); m-fanchiang@nms.ac.jp (M.H.); keikof@nms.ac.jp (K.Y.); 2Division of Breast Surgical Oncology, Department of Surgery, Showa University School of Medicine, Tokyo 142-8666, Japan; ktaruno@med.showa-u.ac.jp (K.T.); or seigonak@gmail.com (S.N.); kuwayama@med.showa-u.ac.jp (T.K.); yoko02410198@yahoo.co.jp (Y.K.); sakashi@med.showa-u.ac.jp (S.A.-T.); 3Tianjin’s Clinical Research Center for Cancer, Tianjin Medical University Cancer Institute and Hospital, Tianjin 300060, China; 4Division of Breast Surgical Oncology, Department of Surgery, Showa University Fujigaoka Hospital, Kanagawa 227-8501, Japan; enotoshi@med.showa-u.ac.jp (K.E.); misakimatsuyanagi@yahoo.co.jp (M.M.); 5Department of Diagnostic Pathology, Nippon Medical School Hospital, Tokyo 113-8603, Japan; takashi-sakatani@nms.ac.jp; 6Center for Data Science, Yokohama City University, Yokohama 236-0027, Japan; sakamaki@yokohama-cu.ac.jp; 7Graduate School of Engineering, University of Tokyo, Tokyo 113-8656, Japan or kuwahata@bee.t.u-tokyo.ac.jp (A.K.); sekino@g.ecc.u-tokyo.ac.jp (M.S.); 8School of Engineering, Tohoku University, Miyagi 980-8579, Japan; 9Graduate School of Agricultural and Life Sciences, University of Tokyo, Tokyo 113-8657, Japan; amkusa@g.ecc.u-tokyo.ac.jp or; 10Department of Research and Development, Matrix Cell Research Institute Inc., Tokyo 101-0025, Japan

**Keywords:** nonpalpable breast lesion, breast cancer, magnetic maker, magnetic probe, surgery

## Abstract

**Simple Summary:**

In this multicenter feasibility study, non-palpable breast lesions in 89 patients were localized using a handheld cordless magnetic probe (TAKUMI) and a magnetic marker (Guiding-Marker System^®^). Additionally, a dye was injected subcutaneously under ultrasound guidance to indicate the extent of the tumor. Consequently, a magnetic marker was detected in all resected specimens, and the initial surgical margin was positive only in five (6.1%) of 82 patients. Thus, the magnetic guiding localization system with ultrasound guidance is useful for the detection and excision of non-palpable breast lesions.

**Abstract:**

Accurate pre-operative localization of nonpalpable lesions plays a pivotal role in guiding breast-conserving surgery (BCS). In this multicenter feasibility study, nonpalpable breast lesions were localized using a handheld magnetic probe (TAKUMI) and a magnetic marker (Guiding-Marker System^®^). The magnetic marker was preoperatively placed within the target lesion under ultrasound or stereo-guidance. Additionally, a dye was injected subcutaneously to indicate the extent of the tumor excision. Surgeons checked for the marker within the lesion using a magnetic probe. The magnetic probe could detect the guiding marker and accurately localize the target lesion intraoperatively. All patients with breast cancer underwent wide excision with a safety margin of ≥5 mm. The presence of the guiding-marker within the resected specimen was the primary outcome and the pathological margin status and re-excision rate were the secondary outcomes. Eighty-seven patients with nonpalpable lesions who underwent BCS, from January to March of 2019 and from January to July of 2020, were recruited. The magnetic marker was detected in all resected specimens. The surgical margin was positive only in 5/82 (6.1%) patients; these patients underwent re-excision. This feasibility study demonstrated that the magnetic guiding localization system is useful for the detection and excision of nonpalpable breast lesions.

## 1. Introduction

The introduction of mammographic screening programs has led to the identification of an increased number of nonpalpable breast lesions. Currently, in developed countries, approximately 20% to 30% of detected breast cancer cases are nonpalpable [1,2]. Moreover, due to the development of neoadjuvant chemotherapy, there has been an increase in the number of patients in whom a complete response was obtained [3]. These lesions are often difficult to identify preoperatively and intraoperatively. Several techniques have been developed for the localization of nonpalpable breast cancers [1,4,5,6,7]. The two most established techniques for pre-operative localization of nonpalpable breast lesions are wire-guided localization (WGL) and radioactive seed localization (RSL). WGL involves the percutaneous implantation of a hooked wire under image guidance to mark the center or outer edges of target lesions. Although WGL is the most commonly used method [8], it has several disadvantages, including mechanical stimulation of wire plucking, kinking, and patient discomfort. On the other hand, RSL involves implanting a small radioactive seed to identify the lesion and/or its borders. RSL overcomes many of the disadvantages of WGL, but it requires a strict nuclear regimen, which is its main limitation for applicability at hospitals.

Recently, a non-wire, non-radioactive localization technique, known as the magnetically guided localization (MGL) method, has been developed as an alternative to WGL and RSL. The MGL method uses a handheld magnetic probe with a cord (Sentimag, Endomagnetics Ltd., Cambridge, England and Wales) and a magnetic marker (Magseed, Endomagnetics Ltd., Cambridge, England and Wales) for localization of nonpalpable lesions [5,9,10,11,12,13]. Currently, the MGL method appears to be a feasible and safe method of breast lesion localization. However, they are commercially not available in Japan. On the other hand, in Japan, a handheld cordless magnetic probe (TAKUMI, Matrix cell Research Institute Inc., Tokyo, Japan) has been developed to detect sentinel lymph nodes (SLNs) in breast cancer patients [14,15]. In the identification rates of SLNs, the MGL method is not inferior to the gamma probe and dye-guided method [15,16,17]. In this study, we evaluated the feasibility of an occult lesion localization technique using the handheld cordless magnetic probe (TAKUMI) and the magnetic marker (Guiding-Marker System^®^, Hakko Co., Ltd., Nagano, Japan). The primary outcome was the successful identification of the guiding marker in the excised specimen. Surgical margin status and re-excision rates were evaluated as the secondary outcomes.

## 2. Materials and Methods

### 2.1. Patients

Patients with nonpalpable breast cancer, who were histologically diagnosed using core needle biopsy (CNB) or vacuum assisted biopsy (VAB), at three hospitals (Nippon Medical School Hospital, Showa University Hospital, and Showa University Fujigaoka Hospital) were enrolled. In this study, two types of magnetic probes (TAKUMI) were used; the first type of magnetic probe was used from January 2019 to March 2019 (Figure 1a: generation 1 (Gen.1)), and the second type of magnetic probe from January 2020 to July 2020 (Figure 1b: generation 2 (Gen.2)).

The female patients, aged ≥20 years, who met the following criteria were included: underwent breast-conserving surgery (BCS) for breast cancer; received neoadjuvant chemotherapy; and underwent tumor excision for indeterminate or suspicious on CNB or VAB. The patients who were pregnant, had inflammatory breast cancer, underwent breast implant insertion, and/or had a metal allergy were excluded. The breast lesion, which could not be palpated during pre-operative examination by the surgeons, was defined as a nonpalpable breast lesion. Written informed consent was obtained from all participants.

### 2.2. Materials

The TAKUMI is a handheld cordless magnetic probe equipped with a permanent magnet and a Hall effect sensor for detecting magnetic objects. The objects are magnetized by the magnetic fields from the permanent magnet, and the newly generated magnetic field from the magnetized objects is detected by the sensor. The value of the detected signal is visible on a small display, and sounds are produced according to the detected values (Figure 1a–c). It was developed at the University of Tokyo under a grant from the Japan Agency for Medical Research and Development. The second type of TAKUMI (Gen.2) is commercially available for sentinel lymph node biopsy and mammary occult lesion localization, which has regulatory approval in Europe for medical device safety (CE marking of conformity, NB:0344, EC certificate No.: 4201663CE01). The second type of TAKUMI (Gen.2) has been improved with the addition of a push-button and easy battery replacement.

The Guiding-Marker System^®^ was used as a magnetic marker. It consists of a stainless-steel hook connected to a 30-cm long 5-0 nylon monofilament suture (Figure 2a). The tip of the marker is bent, and the size of the marker is φ 0.28 mm × 10 mm in length (Figure 2b). The Guiding-Marker System^®^ has been used for thoracoscopic resection of pulmonary nodules [18,19] and for MRI-guided breast lesion mapping [20]. The guiding marker was inserted into the center of the target lesions under ultrasound guidance, using a 21-gauge 10 cm long steel needle. It is similar as the insertion of a conventional breast maker. The flexible nylon suture is not associated with mechanical stimulation of wire plucking, kinking, and patient discomfort seen with WGL.

### 2.3. Surgical Procedures

A 21-gauge 10-cm long steel needle was used to insert the Guiding-Marker System^®^ into the center of the target lesions under ultrasound guidance before surgery (Figure 3a). Only in cases with microcalcification, stereotaxic mammography was used for guidance. The markers were placed the day before surgery, or before surgery under anesthesia. An ultrasound and TAKUMI were used to confirm whether the marker was located within the lesion, and to mark on the skin by ink before incision. The visibility of the puncture needle and the marker was ensured under ultrasound during the procedure (Figure 3a–c). The procedure of the Guiding-Marker System^®^ insertion was performed by experienced breast surgeons.

After sentinel lymph node (SLN) biopsy, a small amount of sterile gentian violet or indigocarmine with gel was injected subcutaneously at several points, at least 5 mm from the edge of the tumor under sonographic guidance, to indicate the ductal spread of the tumor to be excised (Figure 4b).

Then, a skin incision was made over/outside the lesion or areolar line, and TAKUMI in a sterile bag was used to confirm the magnetic marker within the dye-marked area (Figure 4c). Since a magnetic probe reacts to materials containing iron, titanium muscle retractors and surgical equipment were used while checking the position of the lesion intraoperatively. The tumor was resected cylindrically (Figure 4d). Once the specimen was excised, the magnetic probe was used to confirm the presence of a magnetic marker in the resected breast tissue. A radiograph of the specimen was taken to confirm the presence of the lesion and the magnetic marker within the resected specimen (Figure 5). All surgical procedures were performed by experienced breast surgeons.

### 2.4. Pathological Examination

The surgical margin of the resected breast tissue, depending on the physicians’ discretion, was histologically examined on frozen section during the surgery. If the surgical margin was positive for cancer cells on the frozen section, additional breast tissue, corresponding to the positive surgical margin, was resected. On the other hand, a second surgery was performed when the surgical margin was positive for invasive carcinoma on the ink margin on the formalin-fixed paraffin-embedded section. Therefore, the margin status, in most cases, was determined on initially resected breast tissue. Margins were classified based on consensus guidelines [21,22]. For invasive cancer, the margin was considered positive if the tumor was found on the ink margin. For ductal carcinoma in situ (DCIS), the margin was positive if DCIS was present on the ink margin and it was close if DCIS was found within <2 mm.

## 3. Results

Eighty-seven patients were recruited in the study, including 39 patients from January 2019 to March 2019 and 48 patients from January 2020 to July 2020. Eighty-two patients (94.3%) underwent partial mastectomy for breast cancer treatment, five (5.7%) underwent tumor excision for diagnostic purposes. Patient demographics and tumor characteristics are shown in Table 1. The mean age was 54.4 years (range, 33–88 years). Localization markers were placed for masses (71 patients) under sonographic guidance and for calcification (16 patients) under mammographic guidance. Histologically, 64 patients had invasive carcinoma, and 23 had DCIS.

The study outcomes are reported in Table 2. Technical success was defined as the presence of the magnetic marker in the excised specimen on the postoperative specimen radiograph. In 85 out of 87 patients (97.7%), the magnetic markers were detectable on magnetic probing, both during and after the surgical resection of the target lesion. However, in two cases, the magnetic marker, which could not be detected before surgery, was detected intraoperatively. In all cases, guiding markers were removed during the initial surgical operation. The mean size of the lesion was 11.1 mm along the longest dimension, ranging from 0 mm to 33 mm, including non-invasive carcinoma. The size of lesions with pathological complete response after chemotherapy were measured as 0 mm. The mean weight of all excised specimens was 39.9 g, ranging from 2 g to 184 g, depending on the extent of the lesion. Five cases of tumor excision for diagnostic purposes are excluded from the analysis on margins status.

Five (6.1%) of 82 patients, diagnosed with breast cancer on histopathology, were histologically diagnosed with positive surgical margins; four (4.9%) underwent re-excision due to positive surgical margin on the frozen section and one (1.2%) underwent a second surgery due to positive margin on the permanent section. Postoperatively, all patients received breast radiation. Six patients (7.5%) underwent boost radiation therapy because of close margin. There is no difference between the first type (Gen.1) and second type (Gen.2) of TAKUMI in the results of the clinical outcomes. No complications, including allergies and pathological responses to the marker, were observed in marker placement, both preoperatively, intraoperatively, and postoperatively. There are no complaints from patients about discomfort with mechanical stimulation of wire plucking, kinking.

## 4. Discussion

The localization of nonpalpable breast lesions has increasingly become an important component of BCS. The two most established techniques for pre-operative localization of nonpalpable breast lesions are WGL and RSL. However, WGL and RSL, requiring a wire and radioisotope, respectively, have several disadvantages. Hence, the ideal method would be a non-wire, non-radioactive localization method that does not require an energy source. MGL appeared as an effective alternative to WGL and RSL that overcomes the disadvantages of these techniques.

A feasibility study of the MGL technique was performed in the UK [9]. They used Sentimag as a magnetic probe and Magseed as a magnetic tracer. Consequently, all 20 patients with nonpalpable breast cancer underwent successful surgical excision guided by a magnetic probe; however, a surgical re-excision was required in two (10%) patients with positive surgical margins. In a review and pooled analysis of 1559 surgical excisions, Gera et al. [23] reported a successful localization and retrieval rate of 99.9% and a relatively low re-excision rate of 11.3%. Micha et al. [24] reported that the Magseed group obtained the better satisfaction for clinicians in terms of the technical aspects, and also decreased the anxiety of patients between localization and surgery, with comparison to the WGL group. Thus, several studies have demonstrated that MGL using Magseed is an easy, sensitive, and effective localization method [7,8,23,25]. The use of Magseed technology for lesion localization gained US Food and Drug Administration approval in 2016. MGL has been a beneficial addition to the rapidly developing breast localization technologies.

In Japan, neither the magnetic probe (Sentimag) nor the magnetic marker (Magseed) is commercially available. However, a novel handheld cordless magnetic probe (TAKUMI) has been developed to detect the SLNs in breast cancer patients [14,15]. Compared to Sentimag, there is an important difference in operation principle to be realized in the compact shape of TAKUMI. Sentimag utilizes an alternating current (AC) magnetic field generated by an AC power supply. In contrast, TAKUMI utilizes direct current (DC) magnetic fields generated by a permanent magnet without any AC power supply. TAKUMI could be operated with a small battery, therefore, that would make it possible to realize the compact shape and the cordless device.

In BCS, there is a conflict between obtaining an adequate excision margin around the tumor and not removing too much tissue, which may result in breast deformity [26]. Although the MGL technique could detect the guiding marker and give us an accurate localization of the target lesions during the surgery, it could not indicate the extent of the lesion. In several studies evaluating MSL using Magseed, the rate of positive surgical margins ranged from 10% to 16% [16,18,19,20]. These results are not so different from those with WGL (20–70%) [27] and RSL (7–27%) [1,28]. On the other hand, ultrasound guided BCS can overcome the problem of positive surgical margins, although it may miss the location of a small tumor during surgery [29]. Therefore, we verified the MGL method for nonpalpable breast lesions by using a handheld cordless magnetic probe (TAKUMI) and magnetic marker (Guiding-Marker System^®^).

In this study, a dye was injected subcutaneously under ultrasound guidance to indicate the extent of the tumor excision. Consequently, the surgical margin was positive only in 5 (6.1%) of 82 patients, and 6 (7.5%) of 82 patients underwent boost radiation therapy because of close margin. No complication, including allergies and pathological response to the marker, were observed in marker placement preoperatively, intraoperatively, or postoperatively. Thus, the MGL with ultrasound guidance is useful for the detection and excision of nonpalpable breast lesions. However, the present study is a single-arm feasibility study. To further evaluate the usefulness of the magnetic marker and probe method, we would like to conduct a double-blind study, comparing it with conventional methods with a greater sample size and more surgeons.

There are several limitations associated with the use of MGL, including limited use of metal instruments during surgery and limited depth of detection. Because of the magnetic property of the probe, metal instruments containing iron should be kept away from the tip of the detection probe [9]. In this study, titanium muscle retractors and surgical equipment were used while checking the position of the lesion intraoperatively. Recently, a new non-radioactive wireless localization system, called the Magnetic Occult Lesion Localization Instrument (MOLLI), has been developed in the Sunnybrook Odette Cancer Center. The effect of surgical instruments on MOLLI function is minimal, and does not impact its accuracy or reliability [30]. On the other hand, Magseed, which is made of stainless-steel (approximately φ 1 mm × 5 mm in length), can only detect up to a depth of 4 cm, whereas the MOLLI system, which has a custom-made MOLLI marker (φ 1.6 mm × 3.8 mm in length), can detect up to a depth of 53 mm [5]. The Guiding-Marker System^®^ in this study consists of a stainless-steel hook (φ 0.28 mm × 10 mm in length) connected to a 30-cm long 5-0 nylon monofilament suture. In preclinical studies (not published) the Guiding-Marker System^®^ was detectable up to depths of 31 mm when the marker was oriented parallel to the longitudinal axis of the probe. The detectable depth was 21 mm when the marker was oriented perpendicularly. In this study, the magnetic marker could not be detected transcutaneously in 2 cases due to the detectable depth limit, however, these were detected intramammary intraoperatively. The nylon suture helps to avoid losing the marker, so that all the markers were removed during the initial surgical operation. The nylon suture resolved the patient’s discomfort compare to the wire on WGL method. The Guiding-Marker System^®^ was originally developed for the preoperative localization of pulmonary tumors [18], but it may be too shallow to detect a deeply located breast lesion. Technological development of the magnetic marker could resolve this limitation.

## 5. Conclusions

We evaluated the feasibility of an occult lesion localization technique using a handheld cordless magnetic probe (TAKUMI) and a magnetic marker (Guiding-Marker System^®^). The high rate of successful localization and low rate of re-excision support the effectiveness of MGL. The MGL method is a reliable, accurate, and convenient system for localizing nonpalpable breast lesions. This technique does not have disadvantages that are commonly associated with WGL and RGL. However, MGL also has several limitations, such as a limited ability to use surgical metal instruments and limited depth of detection during surgery. Therefore, we need to develop more effective technologies that can, besides localization, determine the extent of the tumor excision.

## Figures and Tables

**Figure 1 cancers-13-02923-f001:**
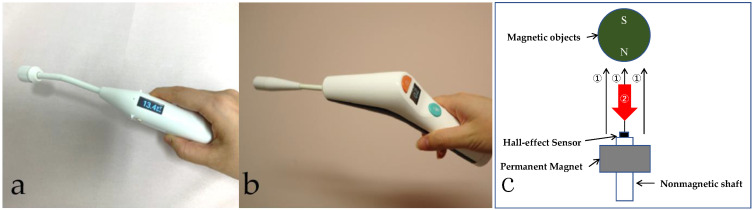
The magnetic probe TAKUMI (Matrix cell Research Institute Inc., Tokyo, Japan): (**a**) First type: Gen.1; (**b**) Second type: Gen.2; (**c**) The detection mechanism of TAKUMI. ① The objects are magnetized by the magnetic fields from the permanent magnet. ② The newly generated magnetic field from the magnetized objects is detected by the Hall effect sensor.

**Figure 2 cancers-13-02923-f002:**
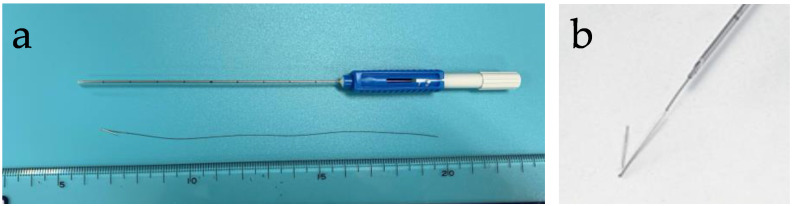
The Guiding-Marker System^®^ (Hakko, Nagano, JAPAN). (**a**) It consists of a stainless-steel hook connected to a nylon thread and a 21-gauge 10 cm long steel needle. (**b**) The tip of the marker is bent, and the size of the marker is φ 0.28 mm × 10 mm.

**Figure 3 cancers-13-02923-f003:**
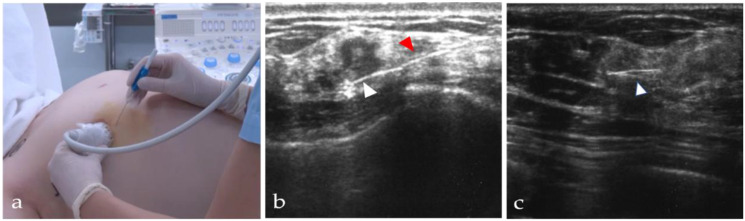
(**a**–**c**) The procedure of the Guiding-Marker System^®^ insertion: (**a**) A needle was inserted using the ultrasound guidance; (**b**,**c**) A needle and a guiding marker have clear visibility under the ultrasound. The white arrowheads in **b** and **c** indicate guiding markers. The red arrowhead in **b** indicates a puncture needle.

**Figure 4 cancers-13-02923-f004:**
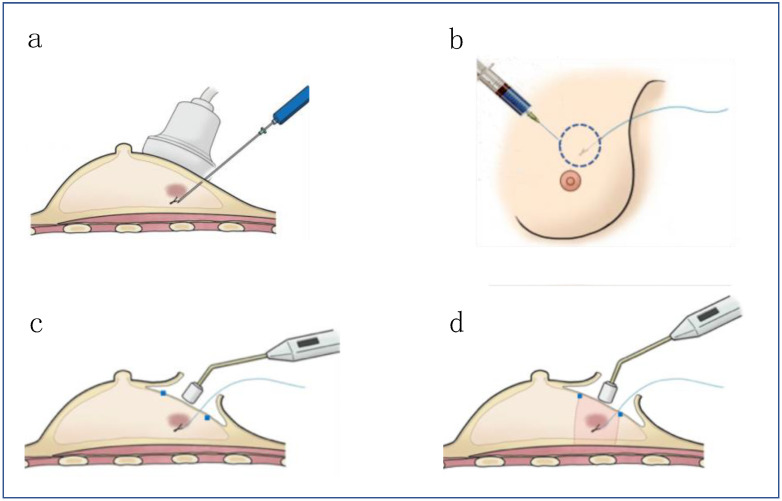
The surgical procedures: (**a**) A marker was inserted into the center of the target lesions using ultrasound guidance; (**b**) A small amount of sterile gentian violet with gel was injected subcutaneously, at least 5 mm from the edge of the tumor under sonographic guidance; (**c**) TAKUMI was used to detect the magnetic marker in the area to be excised; (**d**) The tumor was resected cylindrically.

**Figure 5 cancers-13-02923-f005:**
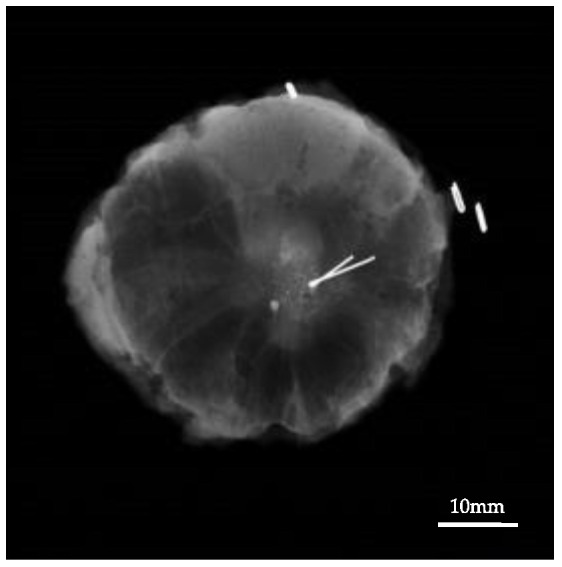
The specimen mammogram showing the calcified lesion and a magnetic marker within the resected specimen.

**Table 1 cancers-13-02923-t001:** Patients and tumor characteristics.

	Gen.1	Gen.2	Total
No. of patients	n = 39	n = 48	n = 87
Ages, y/o (mean)	40–76 (54.4)	33–88 (54.4)	33–88 (54.4)
Menopausal status (Pre/Post/unknown)	14/25/0	22/23/3	36/48/3
Tumor status			
Tumor/low echoic lesion	34 (87.2%)	37 (77.1%)	71 (81.6%)
Microcalcification	5 (12.8%)	11(22.9%)	16 (18.4%)
The size of the lesion			
Clinical tumor size, exclude microcalcification, mm (mean)	0–23 (10.2)	5–30 (11.8)	0–33 (11.1)
Pathological size of lesion, mm (mean)	0–60 (18.4)	0–40 (14.6)	0–60 (16.8)
Histological type of lesion			
Carcinoma in situ	6/39 (15.4%)	17/48 (35.4%)	23/87 (26.4%)
IDC	31/39 (79.5%)	27/48	58/87 (66.6%)
ILC	0/39 (0%)	3/48 (6.3%)	3/87 (3.5%)
Other invasive carcinoma	2/39 (5.1%)	1/48 (2.1%)	3/87 (3.5%)
Histological subtype of breast cancer			
Luminal A	29/39 (74.4%)	41/48 (85.4%)	70/87 (80.5%)
Luminal B	3/39 (7.7%)	4/48 (8.3%)	7/87 (8.0%)
HER2 enriched	3/39 (7.7%)	2/48(4.2%)	5/87 (5.7%)
Triple negative	4/39 (10.2%)	1/48 (2.1%)	5/87(5.7%)
Neoadjuvant chemotherapy	6/39 (15.4%)	3/48 (6.3%)	9/87 (10.3%)
Pathological complete response	3/39 (7.7%)	0/48 (0%)	3/87 (3.5%)

IDC: invasive ductal carcinoma; ILC: invasive lobular carcinoma.

**Table 2 cancers-13-02923-t002:** The results of final pathology and the clinical outcome.

		Gen.1	Gen.2	Total
Surgery	Partial mastectomy	37/39 (94.9%)	43/45 (93.8%)	82/87 (94.3%)
	Tumor excision	2/39 (5.1%)	3/45 (6.2%)	5/87 (5.7%)
Specimen weight, g (mean)		2–131 (38.7)	5–184 (41.9)	2–184 (39.9)
Detectable rate of the Guiding-Marker System^®^	Transcutaneous	38/39 (97.4%)	47/48 (97.9%)	85/87 (97.7%)
	Intramammary	39/39 (100%)	48/48 (100%)	87/87 (100%)
Removal rate of the marker in the specimen		39/39 (100%)	48/48 (100%)	87/87 (100%)
Histological examination of surgical margin	Intraoperative frozen section	7 /37 (18.9%)	9/45 (20.0%)	16 /82 (19.5%)
	Postoperative permanent section	30/37 (82.1%)	36/45 (80.0%)	66/82 (80.4%)
Positive margin status on initial resection	Intraoperative re-excision	3/37 (8.1%)	2 /45(4.4%)	5/82 (6.1%)
	Re-excision on second operation	1/37 (2.7%)	0 /45(0%)	1/82 (1.2%)
Radiation	Whole breast radiation	39/39 (100%)	48/48 (100%)	87/87 (100%)
	Boost radiation	1/37 (2.7%)	5/45 (11.1%)	6/82 (7.5%)

## Data Availability

The data presented in this study are available in this article.

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
