# Peer review of "Magnetically Guided Localization Using a Guiding-Marker System® and a Handheld Magnetic Probe for Nonpalpable Breast Lesions: A Multicenter Feasibility Study in Japan"

_cancers, 2021, doi:10.3390/cancers13122923_

Round 1

Reviewer 1 Report

In the "simple summary" please remove the word precise, this cannot be summarised from this small feasibility study.

In the introduction it would be good to have more information about what the magnetic marker actually is, what its made of, what size it is. how does the magnetic probe work to detect it. A short summary of the technology. how does the technology differ to Magseed or is it the same kind of technology?

In the identification rates of SLNs, the MGL method is not inferior to the gamma 80
probe and dye-guided method. - please provide a reference specific to this statement.

please clarify how many patients had generation one probe and how many generation 2 and comment in the discussion on whether this clinically semed to make any difference.

the primary outcome is the marker being removed (which of course its going to be as its a wire connected to a thread, you are hardly going to leave it in! how many of the index cancers were removed in the same specimen as the marker, i.e. was the marker near the lesion and could be removed together. 

please define what you classify as a close margin in the methods and results section . what width of margin required re-excision??

figure 5 - not sure why this is in this publication as you are showing a graphs result not relevant to the rest of the publication with no methods or validation data attached to this. Maybe better to say that in preclinical studies (not published) the marker was detectable up to depths of ........mm

Please tell us how many were done with generation 1 probe and how many with gen 2, did this make any clinical difference? what was proposed to be better or worse with Gen 2 probe.

do you have a any safety data about complications, allergies, pathological response to marker?

when were the markers placed? how soon in advance of the surgery? on the day of or in advance please?

its not entirely clear to me how lesions were removed? lines 130-136 are causing me confusion - under the heading surggical procedures it says that us was used to confirm marker visibility preoperatively? im not clear what this means. please clarify - did yo place markers and check their placement with us to confirm position then the patient went off to theatre, or was it indeed that during or just before surgery you used US to guide the surgeon to the marker. the key for me is. what was the radiological procedure for placement and checking placement, secondly what was the surgical procedure and how was the surgeon guided onto the marker (ie did the surgeoin ever use and US probe?).

how many of the markers were detectable transcutaneously with the magnetic probe?

so in the discussion i would like a discussion of limitaions of this device - how easy to place, did it move?, did you look at migration? was in palced in advabce? could it be? did patients have a bit of thread hanging out of their breast, what would patients think about this? a comparison of how it is likely to match up against magseed in terms of size of needle, ease of placement, signal depth, ease of surgical use, ease for pathologist cutting up specimen, future studies required, accuracy of device , safety of device, next steps.

please clarify role of ultrasound in this process in the methods results and conclusions, i find it strange that ultrasound underpins a magentic systmen that uses a magnetic probe, this needs discussin in the discussion as to why US is necessary and what would happen without US and any plans to test the device and not use US probes?

Author Response

In the "simple summary" please remove the word precise, this cannot be summarised from this small feasibility study.

⇒removed the word “ precise” in this paper.

In the introduction it would be good to have more information about what the magnetic marker actually is, what its made of, what size it is. how does the magnetic probe work to detect it. A short summary of the technology. how does the technology differ to Magseed or is it the same kind of technology?

⇒Added the description about magnetic marker in “2.2 Materials”

In addition, Guiding-Marker System® was used as a magnetic marker. It consists of a stainless-steel hook connected with a 30-cm long 5-0 nylon monofilament suture (Figure. 2a). The tip of the marker is bent, and the size of the marker is φ 0.28mm x 10mm in length (Figure. 2b).

⇒Added the description about magnetic probe in “2.2 Materials” and “ 4. Discussion”

2.2 Materials

The TAKUMI is a handheld cordless magnetic probe equipped with a permanent magnet and a Hall-effect sensor for detecting magnetic objects. The objects are magnetized by the magnetic fields from the permanent magnet, and the newly generated magnetic field from the magnetized objects is detected by the sensor. The value of the detected signal is visible on a small display, and sounds are produced according to the detected values (Figure. 1a, b, c).

  1. Discussion,

Compared to Sentimag, there is an important difference in operation principle to be realized the compact shape of TAKUMI. Sentimag utilizes AC (alternating current) magnetic field generated by an AC power supply. In contrast, TAKUMI utilizes DC (direct current) magnetic fields generated by a permanent magnet without any AC power supply. TAKUMI could be operated with a small battery, therefore, that would make it possible to realize the compact shape and the cordless device.

In the identification rates of SLNs, the MGL method is not inferior to the gamma 80 probe and dye-guided method. - please provide a reference specific to this statement.

⇒Added [10, 27, 30] at the bottom of this statement. 

please clarify how many patients had generation one probe and how many generation 2 and comment in the discussion on whether this clinically semed to make any difference.

⇒Divided into two group; Gen.1 and Gen.2, and analyzed each of them. There is no clinical difference.

the primary outcome is the marker being removed (which of course its going to be as its a wire connected to a thread, you are hardly going to leave it in! how many of the index cancers were removed in the same specimen as the marker, i.e. was the marker near the lesion and could be removed together.

⇒In all cases, guiding markers were removed during the initial surgical operation.

please define what you classify as a close margin in the methods and results section . what width of margin required re-excision??

⇒Re-excision is required to positive margin with invasive lesion.

figure 5 - not sure why this is in this publication as you are showing a graphs result not relevant to the rest of the publication with no methods or validation data attached to this. Maybe better to say that in preclinical studies (not published) the marker was detectable up to depths of ........mm

⇒Deleted Fig.5 and added below in the sentence.

In preclinical studies (not published) the marker was detectable up to depths of 31 mm when the marker was oriented parallel to the longitudinal axis of the probe. The detectable depth was 21 mm when the marker was oriented perpendicular.

Please tell us how many were done with generation 1 probe and how many with gen 2, did this make any clinical difference? what was proposed to be better or worse with Gen 2 probe.

⇒Divided into two group; Gen.1 and Gen.2, and analyzed each of them. There is no clinical difference.

do you have a any safety data about complications, allergies, pathological response to marker?

⇒I added the sentence below.

No complication including allergies and pathological response to the marker were observed in marker placement, both preoperatively, intraoperatively and postoperatively.

when were the markers placed? how soon in advance of the surgery? on the day of or in advance please?

⇒I changed the sentence. The markers were placed the day before surgery, or before surgery under anesthesia.

its not entirely clear to me how lesions were removed? lines 130-136 are causing me confusion - under the heading surggical procedures it says that us was used to confirm marker visibility preoperatively? im not clear what this means. please clarify - did yo place markers and check their placement with us to confirm position then the patient went off to theatre, or was it indeed that during or just before surgery you used US to guide the surgeon to the marker. the key for me is. what was the radiological procedure for placement and checking placement, secondly what was the surgical procedure and how was the surgeon guided onto the marker (ie did the surgeoin ever use and US probe?).

⇒An ultrasound and TAKUMI were used to confirm whether the marker was located within the lesion, and to mark on the skin by ink before incision. The surgical procedure meant the procedure of the Guiding-Marker System® insertion, it was performed by mostly surgeons in Japan. After incision, we don’t use US, the surgical procedure was performed the guidance of TAKUMI and injected marker of sterile gentian violet with gel, which is injected under US just before surgery. I don’t know this could be the answers what you want…

how many of the markers were detectable transcutaneously with the magnetic probe?

⇒In 2 cases out of 87, it is hard to detect transcutaneous. I fixed the Table 2 to know it.

so in the discussion i would like a discussion of limitaions of this device - how easy to place, did it move?, did you look at migration? was in palced in advabce? could it be? did patients have a bit of thread hanging out of their breast, what would patients think about this? a comparison of how it is likely to match up against magseed in terms of size of needle, ease of placement, signal depth, ease of surgical use, ease for pathologist cutting up specimen, future studies required, accuracy of device , safety of device, next steps.

⇒I added some sentences in this paper.

The marker is placed in advance. No complication including allergies and pathological response to the marker were observed in marker placement, both preoperatively, intraoperatively and postoperatively. The Guiding-Marker System® consists of a stainless-steel hook connected with a 30-cm long 5-0 nylon monofilament suture The nylon suture is resolved the patient’s discomfort compare to the wire on WGL method.

In Japan, both the magnetic probe (Sentimag) and the magnetic marker (Magseed) are not commercially available, so that we are not able to compare each of them.

please clarify role of ultrasound in this process in the methods results and conclusions, i find it strange that ultrasound underpins a magentic systmen that uses a magnetic probe, this needs discussin in the discussion as to why US is necessary and what would happen without US and any plans to test the device and not use US probes?

⇒The MGL technique could detect the guiding marker and give us an accurate localization of the target lesions during the surgery, however, it could not indicate the extent of the lesion. On the other hand, ultrasound-guided BCS, can overcome the problem of positive surgical margins, although it may miss the location of a small tumor during the surgery [23]. Therefore, we verified the MGL method for nonpalpable breast lesions by using a handheld cordless magnetic probe (TAKUMI) and magnetic marker (Guiding-Marker System®).

Reviewer 2 Report

nicely written paper

i do however have some queries. 

1) how does Takumi compare with magseed in term of ease of use, cost and clinical performance?

2) why was there a thread associated with the hook? what is the function of the thread? do the patients complain of the thread as a source of discomfort?

3) when the gentian violet or indigocarmine with gel was injected subcutaneously to mark out the margins of the lesion under ultrasound guidance, did this result in any staining of the skin?

4) was the comestic outcome of the patients assessed as well?

5) was the insertion of the marker done by the surgeons or the radiologists?

6) why was the skin incision made over the lesion and the incision was not concealed at the areola if possible?

7) i do not think that it is fair to include the 5 cases done for diagnostic excision in the analysis for margin since in these cases, a deliberate effort to achieve clear margins might not have been the goal of the operation.

8) was there any reason why the marker was not retrieved in some of the cases?

Author Response

nicely written paper

i do however have some queries.

  • how does Takumi compare with magseed in term of ease of use, cost and clinical performance?

⇒Would be expected the advantages on ease of use and the cost. TAKUMI could be operated with a small battery, therefore, that would make it possible to realize the compact shape and the cordless device. TAKUMI hasn’t had regulatory approval in Japan, and not commercially available so far. The cost would be much cheaper than Magseed, although we are not able to mention it in this paper (around 30 thousand dollars).

  • why was there a thread associated with the hook? what is the function of the thread? do the patients complain of the thread as a source of discomfort?

⇒The nylon suture helps to avoid losing the marker. The nylon suture is resolved the patient’s discomfort compare to the wire on WGL method.

3) when the gentian violet or indigocarmine with gel was injected subcutaneously to mark out the margins of the lesion under ultrasound guidance, did this result in any staining of the skin?

⇒No, it is injected subcutaneously into the mammary gland tissue and has no effect on the skin.

4) was the comestic outcome of the patients assessed as well?

⇒No, we didn’t assessed the cosmetic outcome in this study. Maybe next time…

5) was the insertion of the marker done by the surgeons or the radiologsts?

⇒All surgical procedures were performed by experienced breast surgeons.

All markers were inserted by the surgeon in this study

6) why was the skin incision made over the lesion and the incision was not concealed at the areola if possible?

⇒I changed the sentence.

A skin incision was made over/outside the lesion or areolar line in this study. This method makes it easier to identify lesions, especially in small wounds.

7) i do not think that it is fair to include the 5 cases done for diagnostic excision in the analysis for margin since in these cases, a deliberate effort to achieve clear margins might not have been the goal of the operation.

⇒Thank you for pointed out. I excluded 5 cases for diagnostic excision in the analysis for margin status, and it makes improved the data :) I excluded tumorectomy this time as well in this study.

8) was there any reason why the marker was not retrieved in some of the cases?

⇒In all cases, guiding markers were removed during the initial surgical operation.

Round 2

Reviewer 1 Report

thank you for the improvements to the manuscript, it gives a much clearer idea of the technology

Reviewer 2 Report

satisfactory replies to the queries but there is some language errors which needs to be corrected